# HA-Net: A Lake Water Body Extraction Network Based on Hybrid-Scale Attention and Transfer Learning

**Zhaobin Wang** [1,*], **Xiong Gao** [1] **and Yaonan Zhang** [2]

1   School of Information Science and Engineering, Lanzhou University, Lanzhou 730000, China; xgao2019@lzu.edu.cn
2   National Glaciology Geocryology Desert Data Center, Northwest Institute of Eco-Environment and Resources, Chinese Academy of Sciences, Lanzhou 730000, China; yaonan@lzb.ac.cn
*   Correspondence: wangzhb@lzu.edu.cn

**Abstract:** Due to the large quantity of noise and complex spatial background of the remote sensing images, how to improve the accuracy of semantic segmentation has become a hot topic. Lake water body extraction is crucial for disaster detection, resource utilization, and carbon cycle, etc. The the area of lakes on the Tibetan Plateau has been constantly changing due to the movement of the Earth's crust. Most of the convolutional neural networks used for remote sensing images are based on single-layer features for pixel classification while ignoring the correlation of such features in different layers. In this paper, the two-branch encoder is presented, which is a multiscale structure that combines the features of ResNet-34 with a feature pyramid network. Secondly, adaptive weights are distributed to global information using the hybrid-scale attention block. Finally, PixelShuffle is used to recover the feature maps' resolution, and the densely connected block is used to refine the boundary of the lake water body. Likewise, we transfer the best weights which are saved on the Google dataset to the Landsat-8 dataset to ensure that our proposed method is robust. We validate the superiority of Hybrid-scale Attention Network (HA-Net) on two given datasets, which were created by us using Google and Landsat-8 remote sensing images. (1) On the Google dataset, HA-Net achieves the best performance of all five evaluation metrics with a Mean Intersection over Union (MIoU) of 97.38%, which improves by 1.04% compared with DeepLab V3+, and reduces the training time by about 100 s per epoch. Moreover, the overall accuracy (OA), Recall, True Water Rate (TWR), and False Water Rate (FWR) of HA-Net are 98.88%, 98.03%, 98.24%, and 1.76% respectively. (2) On the Landsat-8 dataset, HA-Net achieves the best overall accuracy and the True Water Rate (TWR) improvement of 2.93% compared to Pre_PSPNet, which proves to be more robust than other advanced models.

**Keywords:** deep convolutional neural network; remote sensing image; semantic segmentation; tibetan plateau; transfer learning; attention mechanism



## 1. Introduction

The Tibetan Plateau is the security barrier of the Asian ecosystem and is also well-known as the world's third pole [1] and Asia's water tower [2]. There are numerous lakes on the Tibetan Plateau, which differ greatly in size and account for the majority of the lake area in China [3]. Lakes have an important research value as the sentinel of environmental change and the signal of climate change [4]. In recent years, it has been shown that global warming accelerates glacial melt [5] and permafrost degradation [6], with most of the lakes expanding in the last 30 years. It can also be confirmed that the lake area of the Tibetan Plateau expands at a pace of 0.83%/year due to increased glacial runoff [7]. The overflow of the salt lakes not only causes pollution of the land and freshwater lakes, which damages the ecological environment, but also disturbs the life of the residents.

More remote sensing satellites, such as multi-spectral satellites, hyper-spectral satellites, and high-resolution satellites, have been developed as global observation technology

and sensor equipment evolve. The obtained remote sensing images not only capture detailed ground information but also allow accurate analysis of interesting regions. As the ability to analyze the amount of information contained in the remote sensing images has gradually grown, so has the need for analysis of the target region. Therefore, the automatic extraction of lake water bodies is important for monitoring the changes of lakes in remote sensing images.

The automatic extraction of water bodies from remotely sensed images is an important part of water resource management and an essential part of remote sensing science. In order to eliminate misleading information such as mountain shadows, cloud shadows, and glacial snow accumulation, traditional methods for water body extraction from a large number of remote sensing images have been proposed, which can be mainly divided into a spectral analysis method based on water indices, and a classification method based on machine learning.

With the increasing demand for massive remote sensing data processing, the water index method is a commonly used method for water extraction, which is mainly designed to enhance water features and suppress non-water features, and then to achieve water extraction by selecting the optimal threshold value. McFeeters et al. [8] used the low reflectance of water in the near-infrared band and the high reflectance in the green band to enhance the features of water bodies, resulting in the Normalised Difference Water Index (NDWI). The proposal of NDWI has contributed to the rapid development of the water extraction field from remote sensing images, with numerous studies carried out by subsequent researchers based on NDWI. In order to solve the problem that NDWI cannot suppress noise well in built-up areas as well as vegetation and soil noise, Xu et al. [9] replaced the near-infrared band with the shortwave-infrared band to enhance the features of open water bodies in remote sensing images by naming it modified NDWI (MNDWI). NDWI has also been used by some researchers in combination with other metrics to remove the interference of shadows on the accuracy of water body extraction. Xie et al. [10] combined NDWI with the morphological shadow index (MSI), which is used to describe shadow areas, to propose NDWI-MSI, which is able to highlight water bodies while suppressing shadow areas. Kaplan et al. [11] proposed the water extraction surface temperature index (WESTI) in combination with NDWI and the surface temperature variability between the water body and other noise to improve the extraction accuracy of water bodies in cold regions.

Although the accuracy of water body extraction based on water index methods is improving, they all require a threshold to distinguish between water and non-water areas, where subjective and static thresholds can lead to over-or underestimation of surface water areas [12]. Due to the advantages of avoiding the manual selection of threshold and better image understanding, the classification methods based on machine learning are often used to extract water bodies from remote sensing images. They mainly use manually-designed waterbody features to form a feature space, which is then fed into a machine learning classifier to achieve water body extraction. Balázs et al. [13] used principal component analysis to reduce the correlation of bands with spectral indices which were fed into a classifier with principal components (PCs) to distinguish between three water-related categories: water bodies, saturated soils, and non-water. Saghafi et al. [14] used the data fusion method to improve the resolution of multispectral images and perform information enhancement, and then used the extracted features to classify high-resolution multispectral images, which demonstrated the significance of multisensor fusion for water body extraction. Although all of the above methods achieve good classification accuracy, they often require a certain amount of a priori knowledge to extract features manually. At the same time, the manually extracted features have certain limitations and lack some generalization ability.

In the special field of computer vision, a deep convolutional neural network (DCNN) has higher accuracy compared to traditional machine learning methods. In 2012, AlexNet [15] was formally proposed for the ImageNet classification task, which established the founda-

tion for the wide application of DCNN. In addition, various backbone networks have been proposed to improve the drawbacks of AlexNet. For example, ResNet [16], ResNeXt [17], and RegNet [18], which use residual learning modules to avoid degradation problems caused by increasing network depth. The lightweight network represented by MobileNet [19] reduces the number of trainable parameters, which divides the general convolution into depthwise convolution and pointwise convolution. Densely-connected convolutional neural networks, such as DenseNet [20], connect the current layer to all previous layers.

The purpose of semantic segmentation is to assign a class label to each pixel based on the probability map calculated by softmax or sigmoid function. DCNNs have achieved superior performance in natural image segmentation, and they are also being used in remote sensing image analysis in recent years [21–24]. However, the complex background of remote sensing images and targets with multi-scale features lead the advanced network not to model the foreground correctly. Therefore, its performance is poor. The remote sensing image analysis is also an important application field of attention mechanism used to minimize noise interference, such as building footprint extraction [25–28], road extraction [29,30], water body extraction [31–33], land cover segmentation [34–37].

Lakes are important indicators of global climate change. Traditional water body extraction methods all suffer from the drawbacks of poor generalization ability, high computational complexity, and low extraction accuracy. Many researchers have used DCNN for the extraction of water bodies. Restricted receptive field deconvolution network (RRF DeconvNet) [21] was proposed to perform accurate extraction of water bodies in remotely sensed images, but without using pre-trained weights to initialize the model. To overcome the boundary-blurring problem of DCNN which is caused by the loss of boundary information during the downsampling process, it proposes a new edge-weighted loss function to assign greater weights to the pixels near the boundary. However, due to the single expansion rate, it is not enough to deal with common problems such as noise interference and multiscale features. Weng et al. [22] improved the feature extraction method by introducing depthwise separable convolution to reduce the risk of overfitting and expanding the receptive field by dilated convolution. The information of the small water body is obtained by using cascading methods in the encoder stage. Guo et al. [23] used four parallel dilated convolutions to form a multiscale feature extractor which guarantees the accuracy of small water body extraction. All the above methods use dilated convolution to capture more multiscale contextual information, however, the shallower encoder cannot extract enough features for pixel classification in the regions with strong noise interference. To reduce the risk of overfitting, Wang et al. [24] used ResNet-101, which introduces depthwise separable convolution, as an encoder to prevent overfitting. Although a high accuracy rate is obtained, its training time is too long.

Attentional mechanisms, an important area of research in the field of computer vision, had been applied by many researchers to the extraction of water bodies from remote sensing images with a focus on the adaptive weighting of different important information. A network of two-branch attention mechanisms [31] was proposed, where a deeper one is used to extract multiscale channel features and another shallower one is used to extract location information, and the two branches were fused to segment the water body. However, it cannot accurately extract small water bodies and edge information. Xia et al. [32] localized the water body by shallow features and a large-scale attention module while segmenting the edges of the water body using deep features and a small-scale attention module, but they used a conditional random field (CRF) to further enhance the extraction capability. Zeng et al. [33] proposed an adaptive row and column self-attention mechanism to achieve high accuracy extraction of aquatic ponds without using post-processing.

In this paper, a model based on a fully convolutional neural network is proposed to perform pixel classification on the Google dataset and validate the robustness of the Landsat-8 dataset. Firstly, we propose a two-branch encoder structure, which uses ResNet-34 to extract deep semantic features of lake water bodies and a feature pyramid network (FPN) to fuse the extracted features of different resolutions. Due to the low interclass

variance and high intraclass variance features, we use hybrid-scales attention block (HAB) to weight the spatial and channel information for feature maps to reduce the interference of noise. During upsampling, the pixelshuffle convolution upsample block (PCUB) is used to recover the low-resolution feature maps to the spatial resolution of the original image and refine the segmentation boundaries. The main contributions of this paper are summarized in the following five points.

1. A Hybrid-scale Attention Network, named HA-Net, is proposed in this paper for the effective extraction of the lake water body.
2. Combining ResNet-34 and FPN, a two-branch encoder structure is proposed to model the foreground, which can better solve the multiscale problem of the lake water body.
3. Inspired by Inception [38], where using convolutional kernels of different sizes can improve the expression ability of the model, we design a hybrid-scale attention block.
4. PCUB, by combining Pixelshuffle [39] and densely-connected convolution, is proposed to perform upsampling, which has better segmentation performance compared to bilinear interpolation upsampling, nearest interpolation upsampling, and transposed convolution upsampling.
5. Compared to other state-of-the-art models, such as DeepLab V3+, PSPNet, Unet, MSLWENet, MWEN, and SR-segnet, HA-Net achieves the best performance on the Google dataset and demonstrates better robustness on the Landsat-8 dataset.

## 2. Related Work

### 2.1. Fully Convolutional Network

Fully convolutional networks(FCN), which use an encoder-decoder structure to recover low-resolution feature maps to high resolution, are used for pixel classification. Long et al. [40] replaced the final fully connected layer of the pure classification network with a decoder that upsamples the feature maps to their original size using transposed convolution, which established the foundation for image segmentation using deep learning methods. To solve the problem of poor accuracy for FCN boundary extraction, a more structured encoder-decoder model, SegNet [41], was proposed, which uses a maximum pooling index in the upsampling phase to introduce more encoding information and recover the resolution of the feature maps. Similar to SegNet, Ronneberger et al. [42] formally proposed the Unet for medical images segmentation which reuses the encoder features by using skip-connection operation during the upsampling process. DeepLab V3+ [43] uses an atrous space pyramidal pooling structure to model multiscale contextual information and an encoder–decoder structure to recover image resolution at the decoder stage. The proposed encoder–decoder architecture has established the foundation for subsequent research work and facilitated the development of the semantic segmentation field.

### 2.2. Attention Mechanisms

The attention mechanism is essentially designed to mimic the way that humans observe objects, being able to dismiss unimportant information while focusing on the most crucial one. Vaswani et al. [44] used the attention mechanism to replace the network structures, such as Recurrent Neural Network (RNN) and Convolutional neural network (CNN) for machine translation, however, it has a quadratic complexity problem and ignores the potential correlation between different samples. In order to model the potential relationships, Zhao et al. [45] introduced one-dimensional convolution with different receptive fields to solve the drawback that self-attention has only global capture capability, but its complexity is still high. Zhai et al. [46] used the product of corresponding positions to eliminate the dot product in self-attention to greatly reduce the computational complexity. In the same year, Guo et al. [47] used two tandem MLPs as memory units, which have linear complexity and implicitly take into account the association of different samples. In addition to the research on self-attention mechanisms, Hu et al. [48] modeled the channel dimension of the feature maps by a squeeze-excitation block, which adaptively calibrates the channel feature response. However, the presence of fully connected layers introduces

a huge amount of parameters and ignores how significant the spatial information is for feature extraction. Woo et al. [49] and Park et al. [50] used tandem and parallel channel attention block and spatial attention block, respectively, for adaptive enhancement of features. However, they ignore the issue that different sizes of convolutional kernels can enhance model expression ability. Li et al. [51] used different sizes of convolutional kernels to obtain feature maps and shared weights of different kernels to improve the training efficiency of the model.

### 2.3. Dilated Convolution

Dilated convolution [52] introduces a new hyperparameter, dilation rate, which defines the distance between each position of the filter. With the decreasing resolution of the feature maps, the receptive field of the convolution kernel becomes larger, but its information also gradually decreases. To improve the accuracy of image classification, He et al. [16] introduced the dilated convolution into ResNet, which is the most widely used backbone network in subsequent research work. Dilated convolution is used to increase the receptive field of the convolution kernels without reducing the resolution of the feature maps, which is very helpful for image segmentation. Chen et al. [53] introduced dilated convolution into the backbone of the network and used a conditional random field instead of the fully connected layer to capture pixel dependencies more accurately for image semantic segmentation. Dilated convolution plays an important role not only in natural image processing but is also frequently employed in remote sensing image analysis [21–24].

### 2.4. Feature Pyramid Networks

In the encoder stage, more detailed information exists in the shallow feature maps to distinguish simple targets, while more semantic information exists in the deep feature maps to distinguish complex targets. For example, VGG [54] only uses the final deep feature maps for classification and ignores the detailed information of other shallow features, which can improve the accuracy of the classification network. Lin et al. [55] fused Shallow detail features and deep semantic features by an FPN, it can better handle the multiscale problem without increasing the computational complexity. FPN can also be used in the semantic segmentation field, Zheng et al. [56] proposed a foreground-aware relation network to enhance the recognition of foreground features and used foreground-aware optimization to balance foreground examples and hard examples during the training process. It can be seen that FPN is a great feature fusion strategy that can be used to model contextual information as well as improve segmentation accuracy.

### 2.5. Global Convolution Network

Semantic segmentation tasks often face two challenges: classification and localization, but they are contradictory. A large number of semantic segmentation models mainly follow the principle of localization and ignore classification. Inspired by the classification model, Peng et al. [57] proposed Global Convolution Network (GCN) and enlarged the size of convolution kernels, which gives the network the same advantages as the classification network to overcome the drawbacks of the semantic segmentation methods that only consider the localization principle. Finally, the segmentation accuracy of the boundary is further improved by adding a boundary refinement block.

### 2.6. Transfer Learning

Semantic segmentation for remotely sensed image analysis has good performance, but it requires learning from a large amount of annotated data. Creating annotated datasets is very expensive, it is crucial to optimize existing datasets [34]. Transfer learning is a great way to solve the problem of model overfitting due to a small dataset, which can be divided into two main aspects. On the one hand, the weights learned on a large dataset are applied to a task with insufficient data to help the model converge to the global optimum values. On the other hand, the model weights obtained in a domain similar to the target task can

be transferred to the target task. This is mainly because some features, such as the shape and color of objects, are relevant and the model weights for these features do not need to be re-learned, only the deep semantic weights need to be fine-tuned. In addition, transfer learning can also be used to validate the robustness of the deep learning methods when applied to different datasets.

## 3. The Proposed Methods

In recent years, deep learning-based supervised methods for natural image processing have greatly contributed to the development of remote sensing image analysis. In this paper, we present the Hybrid-scale Attention Network (HA-Net), which comprises a two-branch encoder, attention block, and decoder to correctly model lake water bodies, with the structure shown in Figure 1. Firstly, we use a two-branch encoder to fuse shallow and deep feature maps without increasing the computational complexity to deal with the multiscale problem of the lake water body. Then, the feature maps outputted from the FPN are fed into the hybrid-scale attention block in four ways, which weigh the channel and spatial information. Finally, we propose PUCB by combining PixelShuffle, and densely connected convolution to capture more precise edge details. The two-branch encoder, hybrid-scale attention block and pixelshuffle convolution upsample block are given in Sections 3.1–3.3, respectively.

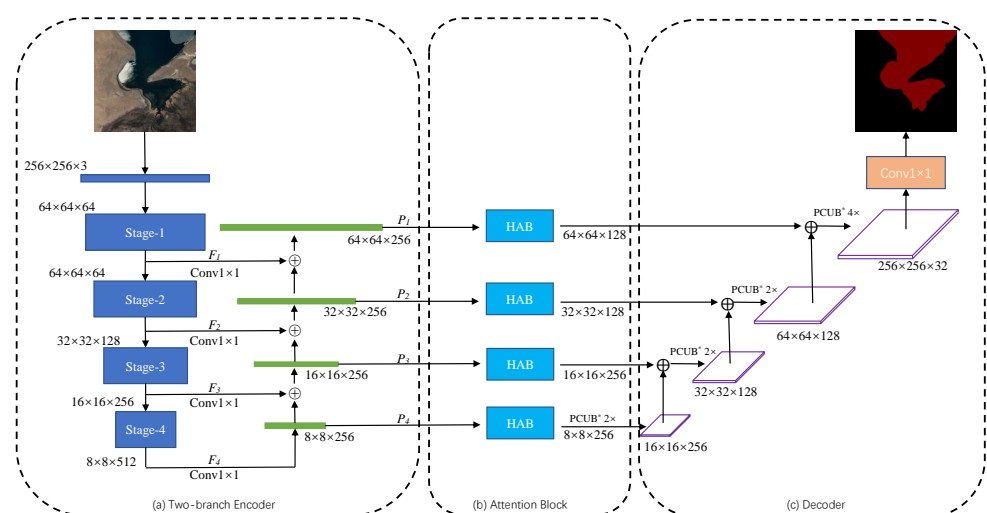

**Figure 1.** The structure of the hybrid-scale attention network used in this paper. (**a**) Two-branch encoder; (**b**) Attention block; (**c**) Decoder.

### 3.1. Two-Branch Encoder

It is well-known that the feature pyramid network is an effective method for fusing multiscale information, which can help us recover details of objects and perform multiscale context modeling. As shown in Figure 1a, a top-down foreground modeling branch and a bottom-up feature fusion branch make up the two-branch encoder. In the top-down branch, we use ResNet-34, a backbone network widely used in image analysis, to extract features from the input training data, where $\{F_i \mid i = 1, 2, 3, 4\}$ indicates the set of extracted feature maps. In the bottom-up branch, the deep feature maps are upsampled to ensure that the spatial and channel dimensions match the shallow feature maps. Pyramid feature maps $\{P_i \mid i = 1, 2, 3, 4\}$, calculated by Equation (1), are generated by element-wise summing of shallow and deep feature maps.

$$P_i = \begin{cases} \zeta F_i + \varsigma P_{i+1} & , i = 1, 2, 3 \\ F_i & , i = 4 \end{cases} \tag{1}$$

where $\zeta$ is the $1 \times 1$ convolution which is used to match the channel dimensions between the shallow and deep feature maps, and $\varsigma$ is the bilinear interpolation upsampling that matches their spatial dimensions.

Due to the high resolution of the shallow feature maps, they contain a significant amount of texture information, while the deep feature maps provide abundant semantic information. The pyramid feature maps are obtained by element-wise summing of shallow and deep features through the bottom-up branch of the two-branch encoder, which can preserve the spatial information of the shallow feature maps and the rich semantic features of the deep feature maps.

### 3.2. Hybrid-Scale Attention Block

In the computer vision field, the attention mechanism uses limited attentional resources to quickly extract the regions that need to be highlighted from the image. Numerous researches have shown that it can be widely used in the fields of video processing, natural language processing, and computer vision. Lots of researches use a single convolutional kernel to extract features, ignoring the multiscale variation problem. Therefore, as shown in Figure 2, we propose a hybrid-scale attention block (HAB) to adaptively weigh the feature maps generated by FPN to suppress noise and background information.

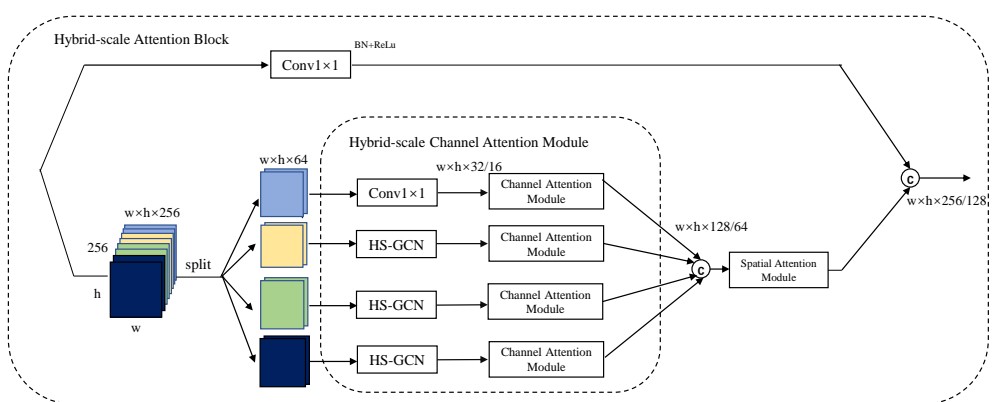

**Figure 2.** The structure of hybrid-scale attention block (HAB).

Inspired by Inception, Hybrid-Scale Global Convolution Network(HS-GCN) is used to improve the expressive capacity of feature maps and weigh the importance of their channel dimensions by using channel attention. Then, the concatenation operation is used on the feature maps that have completed channel attention weighting and weights are assigned adaptively to the spatial dimension. In order to meet the goal of obtaining the information with significant value, HAB dynamically captures the correlation of internal features, while assigning different weights to targets at different scales.

### 3.2.1. Hybrid-Scale GCN

The receptive field is a very important concept for semantic segmentation. For the current end-to-end pixel classification methods, the classifier is locally connected to the feature maps. Therefore, its receptive field can only cover the local part, which is difficult to handle the problem of multiscale variation of the target. Similar to separable convolution, GCN translates $k \times k$ convolution into $1 \times k$ and $k \times 1$ convolution with fewer parameters while ensuring the same receptive field.

The GCN-based semantic segmentation model achieves better results in image segmentation tasks [34,57,58]. We improve the original GCN, as shown in Figure 3, from a single-size convolutional kernel to a two-by-two combination of three convolutional kernels with different sizes, $3 \times 3$, $5 \times 5$, and $7 \times 7$. We add Batch Normalization (BN) [59] and Rectified Linear Unit (ReLU) [60] to each branch of the GCN to reduce the interdependence

of parameters, and finally change the strategy of features fusion. Its formula is shown as follows.

$$Z = \Gamma(Z_{w \times h \times c} SC_{k \times k}, Z_{w \times h \times c} SC_{s \times s}), \tag{2}$$

where $Z_{w \times h \times c}$ represents the input feature map with width $w$, height $h$ and number of channels $c$, $SC$ is a separable convolution with convolution kernel size $k$ or $s$, and $\Gamma(\cdot)$ represents the channel concatation operation.

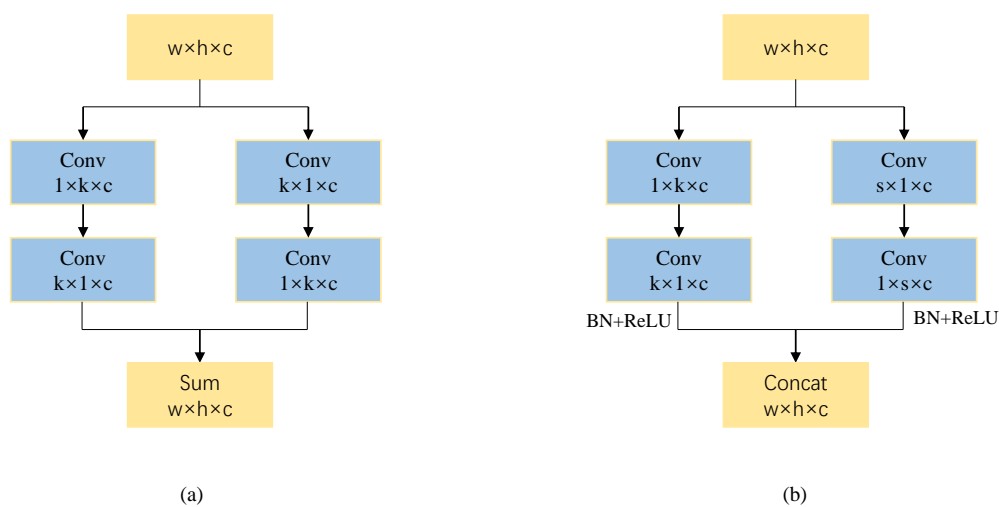

(a)　　　　　　　　　　　　　　　　　　　　　　　　　(b)

**Figure 3.** The differences between GCN and Hybrid-scale GCN. (**a**) GCN; (**b**) Hybrid-scale GCN.

### 3.2.2. Channel Attention Module and Spatial Attention Module

The attention module in the computer vision field can achieve the redistribution of resources and improve the accuracy of lake water body extraction. In this paper, channel attention and spatial attention mechanisms, with the structure shown in Figure 4, are used to adaptively assign weights to the captured water body characteristics.

In the channel attention module, we generate the statistical vector $Z_{ch}$ by compressing the input feature maps $Z_{in}$ in spatial dimension $H \times W$, which is calculated by Equation (3).

$$Z_{ch} = \frac{1}{H \times W} \sum_{i=1}^{H} \sum_{j=1}^{W} Z_{in}(i, j). \tag{3}$$

In this paper, the global average pooling and global maximum pooling are used to generate the channel statistical information $Z_{ch\_avg}$ and $Z_{ch\_max}$, respectively. Then, multilayer perceptron (MLP) is used to efficiently combine the linear information between high and low channels, and the formula is shown below.

$$Z_{mlp} = F(Z_{ch}; W) = W_2(\delta(W_1 Z_{ch})), \tag{4}$$

where $W$ and $\delta$ represent the fully connected layer and ReLU function, respectively. After obtaining the statistical information $z_{mlp\_avg}$ and $z_{mlp\_max}$ by MLP, the probability prediction matrix, which is the importance of each channel, can be obtained by element-wise summing and passing through the sigmoid function. Therefore, the feature maps $Z_{out}$ weighted by the channel attention module can be calculated by Equation (5).

$$Z_{out\_ch} = Z_{in} \times \sigma\left(Z_{mlp\_avg} + Z_{mlp\_max}\right), \tag{5}$$

where $\sigma$ is the sigmoid function.

In the spatial attention module, we compress the feature maps $Z_{ch\_concat}$, which have passed through the channel attention module, by averaging pooling and maximum pooling

in channel dimension $C$ to obtain the feature maps $Z_{sp\_avg}$ and $Z_{sp\_max}$. The formula is shown as follows.

$$Z_{sp} = \frac{1}{C} \sum_{c=1}^{C} Z_{ch\_concat}(c). \tag{6}$$

Then, they are concat at channel dimension and passed through a $1 \times 1$ convolution to generate a feature descriptor. Finally, the probability prediction matrix generated by a sigmoid function is element-wise multiplied with $Z_{ch\_concat}$ to obtain the weighted feature maps, which is calculated by Equation (7).

$$Z_{out\_sp} = Z_{ch\_concat} \times \sigma\big(W_{1\times1} \cdot \Gamma(Z_{sp\_avg}; Z_{sp\_max})\big). \tag{7}$$

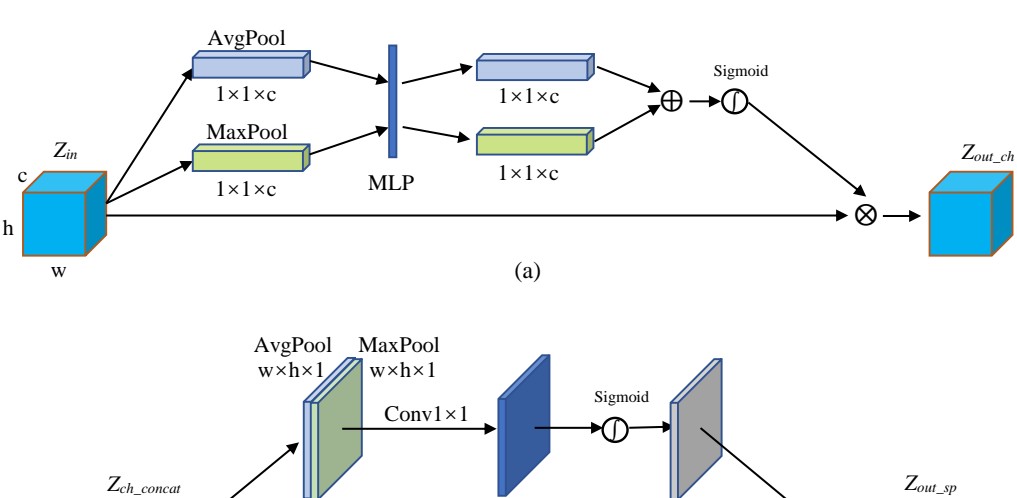

**Figure 4.** The structure of attention modules. (**a**) Channel attention module; (**b**) spatial attention module.

### 3.3. Pixelshuffle Convolution Upsample Block

The spatial resolution of the feature maps weighted through HAB is recovered by the pixelshuffle convolutional upsampling block (PCUB), which is shown in Figure 5. For recovering the spatial resolution of low-resolution feature maps and improve efficiency, we first use sub_pixel convolution in Pixelshuffle [39] with a convolution stride of $1/r$, which changes its spatial resolution to $r$ time of the original feature maps. Then, $1 \times 1$ convolution followed by Batch Normalization and ReLU layers are used to match the channel dimension of the feature maps and increase their nonlinearity. Finally, we model the boundary refinement block as a densely connected structure. More specifically, we define $\tilde{F}$ as the score graph refined by the boundary refinement block: $\tilde{F} = BR(F)$, where $F$ and $BR(\cdot)$ present the rough score graph and the dense connected structure, respectively.

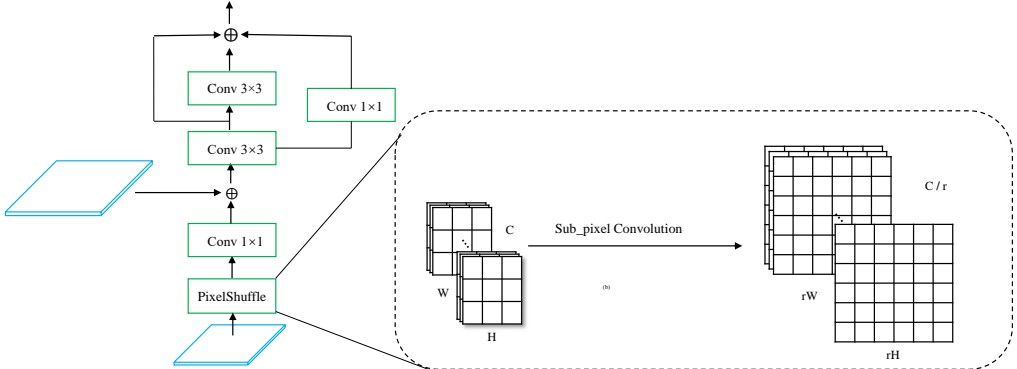

**Figure 5.** The structure of pixelshuffle convolution upsample block.

## 4. Experiment

### 4.1. Datasets

#### 4.1.1. Google Dataset

Google Earth is a virtual earth software developed by Google which is a combination of satellite images and aerial photography data. In this paper, we randomly counted 6164 images for the training set and the remaining 610 images for the test set. This dataset has a total of 6774 RGB images and corresponding RGB ground truth labels (red for the lake water body, black for the background) with a resolution of $256 \times 256$, which were annotated by the deep learning image annotation software, labelme. In this paper, we randomly counted 6164 images for the training set and the remaining 610 images for the test set. Due to the small size of our dataset, we did not set up a validation set.

#### 4.1.2. Landsat-8 Dataset

The Operational Land Imager (OLI) and Thermal Infrared Sensor (TIRS) are carried by Landsat-8, the eighth Earth observation satellite launched by the American Landsat program, and it will provide multi-spectral image data with medium resolution (30 m spatial resolution) for at least eight years. The spatial resolution of Landsat-8 images can be increased to 15 m by fusing multispectral bands with the panchromatic band (band 8). As shown in Figure 6, we selected four Landsat-8 remote sensing images with a large number of lakes in different seasons. Then, ENVI, the remote sensing image processing software, was used for radiometric calibration of the multispectral and the panchromatic bands. Finally, we used the Cubic Convolution in Gram-Schmidt Pan Sharpening to fuse their R, G, and B bands with the panchromatic band to increase the spatial resolution. To ensure that the proposed method, HA-Net, is robust, the original images are divided into non-overlapping patches, and 841 RGB images are randomly selected to form the Landsat-8 dataset. As with the Google dataset, the ground truth labels are also labeled by labelme, with the same image size and format.

#### 4.1.3. Data Augmentation

Although deep learning methods can automatically learn features at different scales from training data to obtain great prediction performance. However, in a specific field, it is so difficult to obtain the labeled ground truth (GT) labels that the weights of the model cannot converge to the global optimum values, thus limiting the working performance. Data augmentation is an important method which mainly used to generate new samples to compensate for the small training datasets. All training samples in the Google dataset are randomly flipped and rotated at any angle of $(0°, 90°)$ before each iteration, as shown in Figure 7, to enhance the generalization of the model.

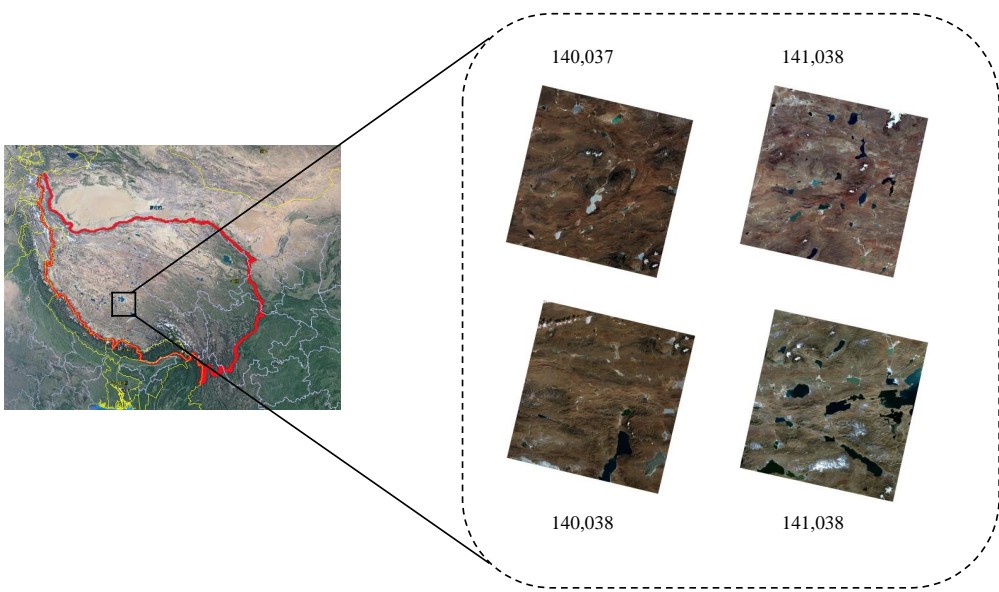

**Figure 6.** The location of images in the Landsat-8 dataset.

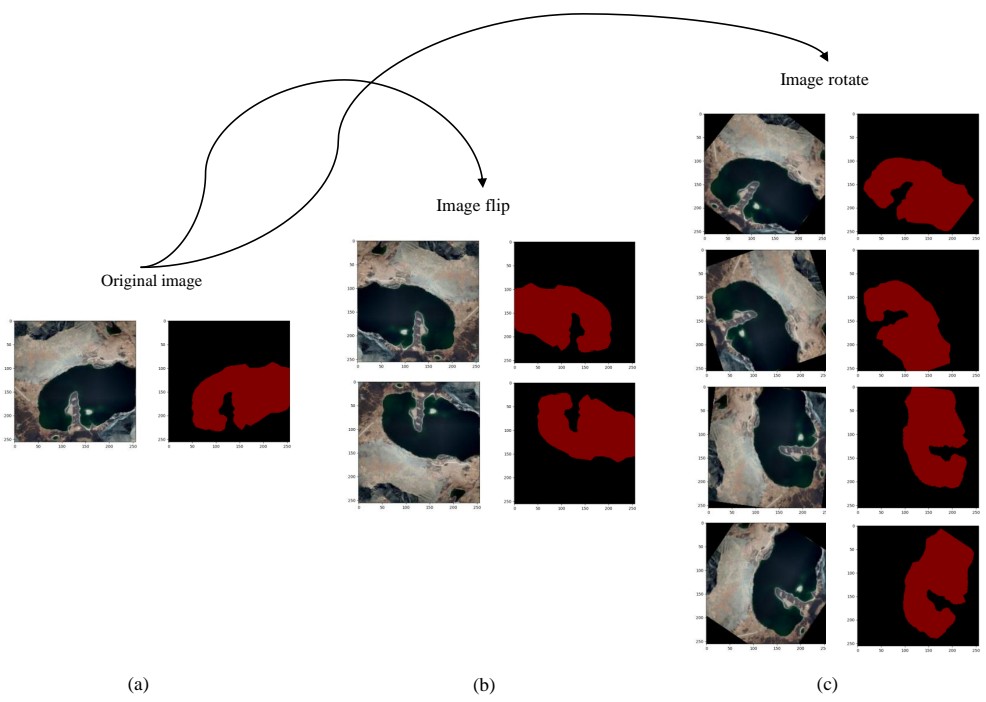

**Figure 7.** Part of the results for data augmentation on the original image and GT label. (**a**) Original image; (**b**) Image flip up to down and left to right; (**c**) Image rotate at any angle of $(0°, 90°)$.

### 4.2. Experimental Details and Evaluation Metrics

4.2.1. The Training Procedure of the HA-Net

Since the Google training set only contains 6164 images, which tends to make the parameters of the network fall into local minima, the transfer learning strategy is used to train the HA-Net. As shown in Figure 8, our model training process is divided into two stages. In Stage-1, we use ResNet-34 with pre-trained weights as a multiscale feature extractor in the two-branch encoder, and all other trainable parameters are initialized randomly with KaiMing initialization [61]. In the forward propagation process, the prediction maps and the GT labels are put through the loss function to calculate the loss values. The parameters of the ResNet-34 are frozen during the backpropagation phase, while other parameters are

updated using the gradient descent method. In Stage-2, all parameters of the model are fine-tuned using a smaller learning rate.

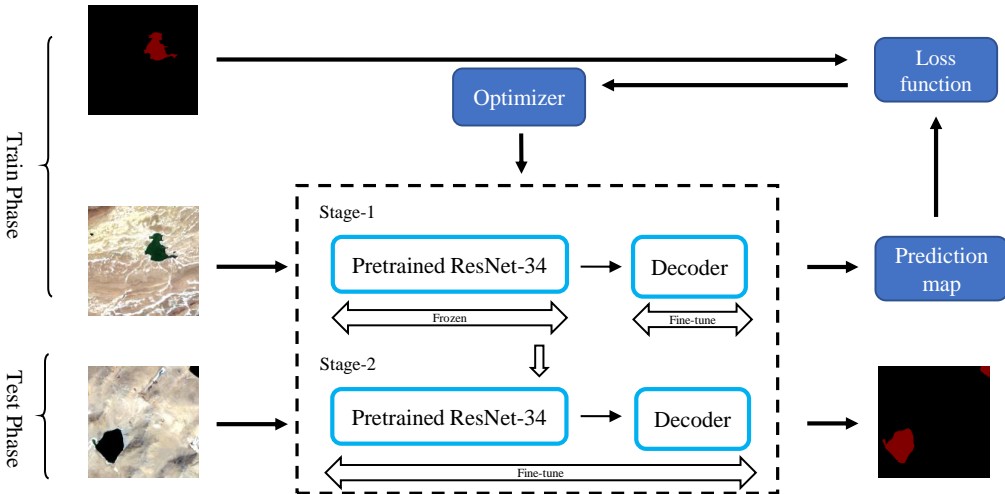

**Figure 8.** The training process proposed in this paper.

4.2.2. Experimental Settings

In this work, the proposed network, HA-Net, was implemented using the open-source framework PyTorch [62]. The code written by Python3.6 was performed on the Windows 10 platform with an NVIDIA 2070 SUPER GPU (8 GB RAM) and AMD Ryzen 5 3600 CPU (3.6 GHz). The loss values between the probability prediction maps and GT labels were calculated using the cross-entropy loss function [63] during the training phase. The model was trained with an Adam optimizer [64], where weight decay was 0.0001. For Stage-1, the initial learning rate and training epoch were set to 0.001 and 20, respectively. For Stage-2, the initial learning rate was set to 0.0001 to fine-tune all parameters and the weights converged to the optimum after 35 epochs. The batch size (the number of samples input to the model at the same time in each training iteration) was 8, and the learning rate was dynamically updated by Equation (8).

$$lr = lr_0 \times 0.8^{(epoch+1)/2} \tag{8}$$

where $lr_0$ is the initial learning rate. For the other comparison models, we used the same parameter settings and retrained them on the Google dataset.

4.2.3. Evaluation Metrics

In general, pixel-level evaluation metrics, which compute some relationship between correctly and incorrectly classified pixels on a pixel-by-pixel basis, are often used to measure the performance of semantic segmentation models. The method of extracting lake water bodies is different from classifying the image itself, which is ultimately a pixel classification method. Therefore, the following evaluation metrics for pixel classification are used to assess the performance of the advanced model.

- Overall Accuracy (*OA*): the proportion of pixels that are correctly classified among the total number of pixels.

$$OA = \frac{TP + TN}{TP + TN + FP + FN} \times 100\% \tag{9}$$

- *Recall*: the proportion of pixels correctly classified as a water body among all water body pixels

$$Recall = \frac{TP}{TP + FN} \times 100\% \tag{10}$$

- Mean Intersection over Union (*MIoU*): The average value of the ratio between the intersection and the concatenation for all classes.

$$MIoU = \frac{1}{k+1} \sum_{i=0}^{k} \frac{TP}{FN + TP + FP} \times 100\% \qquad (11)$$

- True Water Rate (*TWR*): the proportion of pixels correctly classified as a water body among all pixels predicted to be a water body.

$$TWR = \frac{TP}{TP + FP} \times 100\% \qquad (12)$$

- False Water Rate (*FWR*): the proportion of pixels incorrectly classified as a water body among all pixels predicted to be a water body.

$$TWR = \frac{FP}{TP + FP} \times 100\% \qquad (13)$$

where we assume that there are $k$ categories (including one invalid category—background), $TP$, $FP$, $TN$ and $FN$ are true positive, false positive, true negative and false negative, respectively. In order to assess the statistical significance of the different methods, we conducted five experiments using bootstrap analysis, with 6164 samples selected at random with a put-back for each experiment and calculated $p$-value for MIoU using paired t-tests.

*4.3. Experimental Results on Google Dataset*

4.3.1. Performance Comparison of CNN Models

To analyze the effectiveness of HA-Net, we have compared it with other end-to-end advanced semantic segmentation models. Among them, DeepLab V3+ and PSPNet achieved excellent results on natural image segmentation datasets, and Unet established the foundation for the medical semantic segmentation methods. The continuous proposal of semantic segmentation methods for natural image segmentation has promoted the development of remote sensing image analysis. When existing models are directly applied to analyze remote sensing images, they face significant challenges such as the diversity of spatial target distributions and a large amount of noise interference. Therefore, advanced natural image semantic segmentation methods are not able to identify the target correctly. MSLWENet, MWEN, and SR-segnet are networks that are specifically used to extract water bodies from remotely sensed images.

We use the above five evaluation metrics to assess the performance of the models, and the results are shown in Table 1. For DeepLab V3+, PSPNet, and Unet with pre-trained weights, the models with deep encoders tend to achieve better results. HA-Net has a great performance improvement compared to Pre_DeepLab V3+, the best natural image segmentation method, especially the MIoU is improved by 1.04%. For the water body extraction semantic segmentation method, HA-Net improves the OA and MIoU by 0.35% and 1.29% relative to MSLWENet, respectively, and achieves the best performance on the Google dataset for all five evaluation metrics.

The number of trainable parameters and the training time is shown in Table 2. Although the SR-segnet model has the fewest parameters, its training time is not reduced due to the introduction of numerous depthwise separable convolutions. Our proposed model has the least training time and achieves a balance of accuracy and efficiency.

**Table 1.** The quantitative results of comparing the proposed method against other methods.

| Methods | OA | Recall | MIoU | TWR | FWR | *p*-Value (MIoU) |
|---------|-----|--------|------|-----|-----|------------------|
| HA-Net | **98.88%** | **98.03%** | **97.38%** | **98.24%** | **1.76%** | - |
| MSLWENet | 98.53% | 97.67% | 96.09% | 97.47% | 2.53% | $6.45 \times 10^{-9}$ |
| MWEN | 97.75% | 96.94% | 94.84% | 96.94% | 3.06% | $3.05 \times 10^{-13}$ |
| SR-segnet | 97.59% | 95.71% | 94.46% | 96.30% | 3.70% | $2.83 \times 10^{-11}$ |
| Pre_DeepLab V3+ | 98.42% | 97.31% | 96.34% | 97.46% | 2.54% | $5.03 \times 10^{-11}$ |
| Pre_PSPNet | 98.33% | 97.11% | 96.13% | 97.36% | 2.54% | $2.01 \times 10^{-12}$ |
| Pre_Unet | 97.42% | 94.69% | 94.12% | 96.87% | 3.13% | $3.59 \times 10^{-15}$ |

**Table 2.** The comparison results for model complexity and training time.

| Methods | Number of Trainable Parameters | Training Time (s) |
|---------|-------------------------------|-------------------|
| HA-Net | $2.41 \times 10^{7}$ | **156.3** |
| MSLWENet | $2.90 \times 10^{7}$ | 254.8 |
| MWEN | $2.89 \times 10^{7}$ | 161.6 |
| SR-segnet | **$1.35 \times 10^{7}$** | 157.6 |
| Pre_DeepLab V3+ | $5.89 \times 10^{7}$ | 255.0 |
| Pre_PSPNet | $6.76 \times 10^{7}$ | 375.8 |
| Pre_Unet | $3.10 \times 10^{7}$ | 221.1 |

4.3.2. Performance Comparison in Small Lake and Noisy Regions

The Tibetan Plateau lakes can be grouped into small water regions, low interclass variance regions, and high intraclass variance regions [24]. In this section, some visible results, as shown in Figure 9, are selected to demonstrate that HA-Net is superior in extracting small lake water bodies as well as solving the problem of noise disturbance. Both HA-Net and MSLWENet can extract small lake water bodies and solve the interference of noise well, and the performance difference is difficult to distinguish, so we perform quantitative calculations on the images in Figure 9, which are shown in Table 3.

It is clear from the quantitative results that our model improves the OA, Recall, MIoU, and TWR by 0.37%, 0.38%, 0.75%, and 0.55% respectively compared to MSLWENet. Although MSLWENet is able to locate the lake water body well and solve the interference of noise, the boundary extraction of HA-Net is more accurate. As shown in Figure 9(1), for a wide variety of lakes, HA-Net can correctly identify them, and the segmentation boundary is better than other methods. In the noisy regions, as shown in Figure 9(2,3), HA-Net and the models with deep encoders, such as MSLWENet, Pre_DeepLab V3+, and Pre_PSPNet, can handle the interference of noise, while the segmentation results of other models have a large number of mis-segmentations. As shown in Figure 9(4,5), the small lake water body can be basically located correctly, but the segmentation boundaries have large mistakes. Pre_DeepLab V3+, Pre_PSPNet, and Pre_Unet both use pre-trained weights that can help converge to optimal values, but there are still major difficulties in applying them to lake water body extraction. MWFE and SR-segnet cannot extract enough features for pixel classification due to the use of shallow encoders.

**Table 3.** The quantitative results of comparing the proposed method against other methods within the region of Figure 9.

| Methods | OA | Recall | MIoU | TWR | FWR | *p*-Value (MIoU) |
|---|---|---|---|---|---|---|
| HA-Net | **98.21%** | 97.77% | **96.37%** | **97.84%** | **2.16%** | - |
| MSLWENet | 97.84% | 97.39% | 95.62% | 97.29% | 2.71% | $3.77 \times 10^{-7}$ |
| MWEN | 93.85% | 97.52% | 87.84% | 87.11% | 12.89% | $6.42 \times 10^{-17}$ |
| SR-segnet | 94.90% | 97.49% | 89.86% | 89.79% | 10.21% | $9.24 \times 10^{-16}$ |
| Pre_DeepLab V3+ | 97.19% | **97.87%** | 94.34% | 95.19% | 4.81% | $6.48 \times 10^{-11}$ |
| Pre_PSPNet | 96.98% | 95.97% | 93.95% | 96.65% | 3.35% | $1.06 \times 10^{-12}$ |
| Pre_Unet | 95.40% | 94.48% | 90.91% | 94.20% | 5.80% | $1.33 \times 10^{-14}$ |

### 4.3.3. Performance Comparison in Boundary Regions

In this section, the performance comparison of the proposed model with MSLWENet, SR_segnet, MWEN, Pre_DeepLab V3+, Pre_PSPNet, and Pre_Unet on the accuracy of the extracted boundaries will be discussed, which are visualized in Figure 10. In Figure 10(1), HA-Net, MSLEWENet, SR-segnet, and Pre_Unet can correctly localize the slender water body, but the segmentation boundary of SR-segnet and MSLEWENet is not accurate, and Pre_Unet cannot suppress the interference of noise well. In others, all models can correctly localize the water body, but HA-Net has a clear advantage in pixel classification with boundaries. The benefit of the strategy proposed in Sections 3.2 and 3.3 makes HA-Net share the same benefits as the pure classification model to achieve the correct classification of water body edge pixels. The quantitative results are shown in Table 4, where our proposed model obtained the optimum in all five evaluation metrics.

**Table 4.** The quantitative results of comparing the proposed method against other methods within the region of Figure 10.

| Methods | OA | Recall | MIoU | TWR | FWR | *p*-Value (MIoU) |
|---|---|---|---|---|---|---|
| HA-Net | **97.88%** | **96.27%** | **95.64%** | **98.28%** | **1.72%** | - |
| MSLWENet | 97.45% | 95.77% | 94.78% | 97.68% | 2.32% | $1.33 \times 10^{-8}$ |
| MWEN | 96.93% | 95.64% | 93.73% | 96.38% | 3.62% | $6.82 \times 10^{-11}$ |
| SR-segnet | 96.57% | 95.41% | 93.02% | 95.65% | 4.35% | $3.88 \times 10^{-11}$ |
| Pre_DeepLab V3+ | 97.20% | 94.85% | 94.29% | 98.02% | 1.89% | $1.21 \times 10^{-10}$ |
| Pre_PSPNet | 96.98% | 95.97% | 93.95% | 96.65% | 3.35% | $1.66 \times 10^{-10}$ |
| Pre_Unet | 96.39% | 95.13% | 92.67% | 95.48% | 4.52% | $1.45 \times 10^{-12}$ |

### 4.3.4. Performance Comparison of the Proposed Method with Different Encoders

The encoder, which is a very important part of the semantic segmentation network, is used to extract the features of the image. As shown in Table 5, the performance of the model does not consistently increase as the depth of ResNet increases. This is mainly because complex models suffer from the risk of overfitting, which causes a decrease in the overall accuracy. Each layer of DenseNet-121 can directly obtain the gradient and the original input signal, which leads to a kind of implicit deep supervision, and should have better performance in theory compared to ResNet. However, due to the presence of dilated convolution in ResNet, which can expand the receptive field of the convolution kernel, so it has better performance than DenseNet. EfficientNet-b4 reconsiders the model performance in three perspectives of depth, width, and image resolution, resulting in good model performance but with poor efficiency. In addition, the model with VGG-16 as the feature encoder achieved the worst accuracy. So in this paper, we use the most commonly used ResNet-34 as our feature encoder.

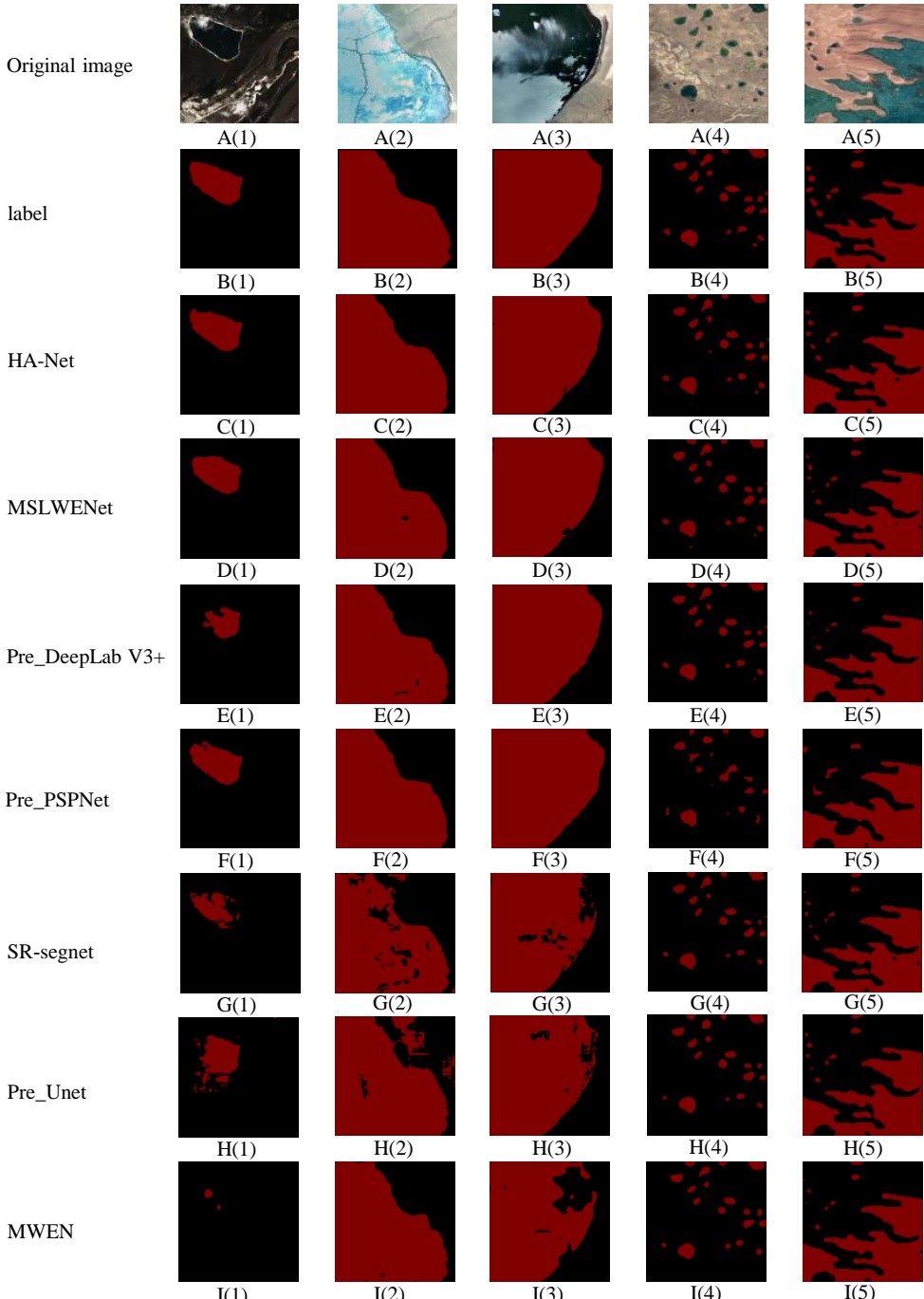

**Figure 9.** The visualization results of comparing the proposed method against other methods in small lake and noisy regions. (**A(1)**–**A(5)**) are the raw images in the Google test set; (**B(1)**–**B(5)**) are the ground truths to which the raw images correspond; (**C(1)**–**C(5)**), (**D(1)**–**D(5)**), (**E(1)**–**E(5)**), (**F(1)**–**F(5)**), (**G(1)**–**G(5)**), (**H(1)**–**H(5)**), and (**I(1)**–**I(5)**) are the extracted results of HA-Net, MSLWENet, Pre_DeepLab V3+, Pre_PSPNet, SR-segnet, Pre_Unet, and MWEN, respectively.

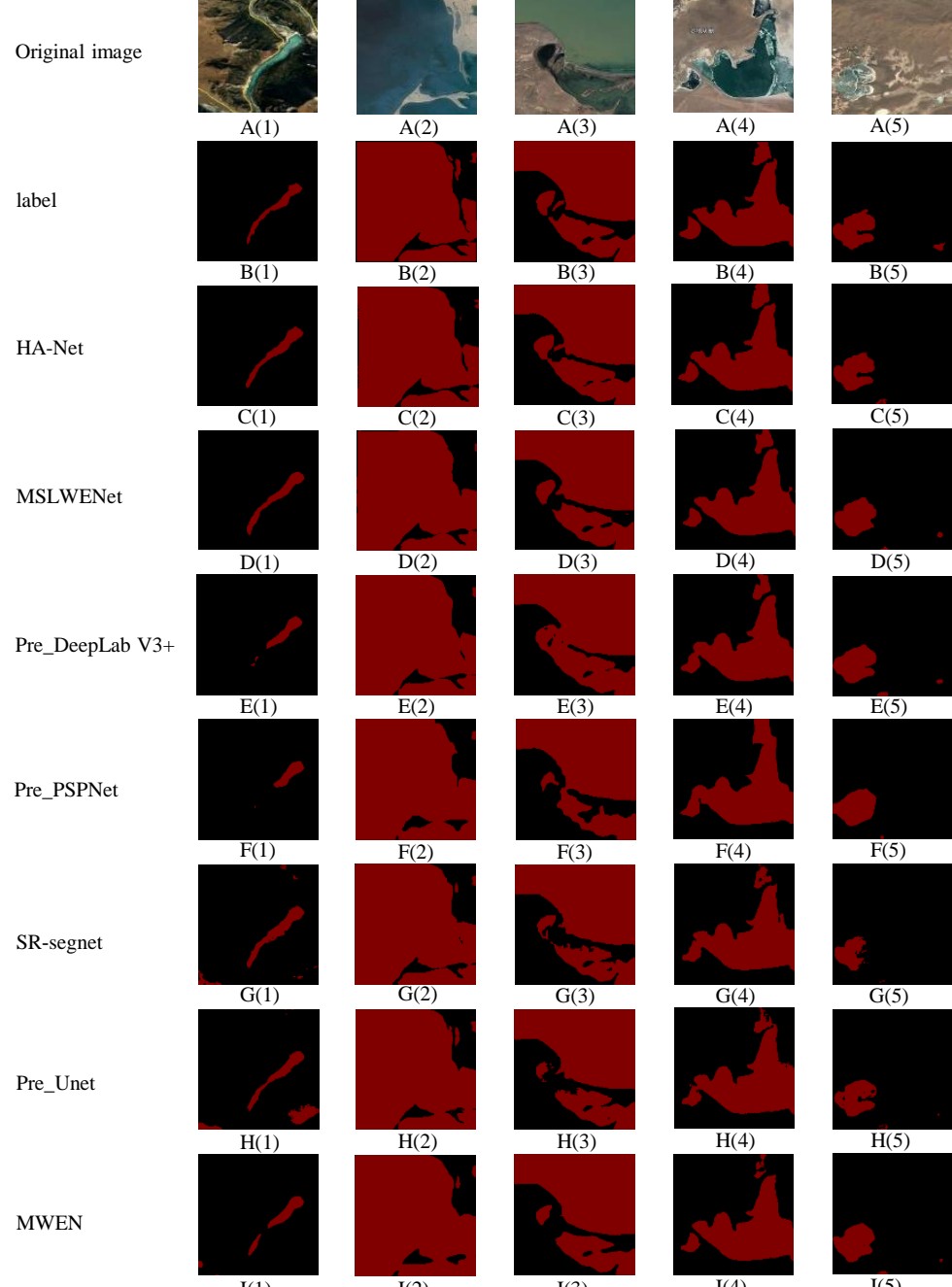

**Figure 10.** The visualization results of comparing the proposed method against other methods in boundary regions. (**A(1)**–**A(5)**) are the raw images in the test set; (**B(1)**–**B(5)**) are the ground truths to which the raw images correspond; (**C(1)**–**C(5)**), (**D(1)**–**D(5)**), (**E(1)**–**E(5)**), (**F(1)**–**F(5)**), (**G(1)**–**G(5)**), (**H(1)**–**H(5)**), and (**I(1)**–**I(5)**) are the extracted results of HA-Net, MSLWENet, Pre_DeepLab V3+, Pre_PSPNet, SR-segnet, Pre_Unet, and MWEN, respectively.

**Table 5.** The quantitative results of comparing the proposed method with different encoders.

| Encoder | OA | Recall | MIoU | TWR | FWR | *p*-Value (MIoU) |
|---|---|---|---|---|---|---|
| ResNet-34 | **98.88%** | **98.03%** | **97.38%** | **98.24%** | **1.76%** | - |
| ResNet-101 | 98.86% | **98.03%** | 97.33% | 98.18% | 1.82% | $8.48 \times 10^{-3}$ |
| VGG-16 | 98.76% | 97.70% | 97.10% | 98.18% | 1.82% | $4.57 \times 10^{-4}$ |
| DenseNet-121 | 98.75% | 97.71% | 97.07% | 98.13% | 1.87% | $1.26 \times 10^{-6}$ |
| EfficientNet-b4 | 98.83% | 97.90% | 97.27% | 98.22% | 1.78% | $3.90 \times 10^{-4}$ |

4.3.5. Ablation Experiments of Attention Blocks and Different Upsampling Methods

Our strategy aims to increase the scores of different evaluation metrics by using spatial attention, channel attention at different scales for adaptive weighting of the feature maps, in addition to using PUCB to eliminate some artifacts in the upsampling process. We perform ablation experiments on HA-Net, where HA-Net-T1 represents the elimination of the Hybrid-scale channel attention block, HA-Net-T2 represents the elimination of the spatial attention block, HA-Net-T3 represents the use of bilinear interpolation upsampling, HA-Net-T4 represents the use of nearest interpolation upsampling, and HA-Net-T5 represents upsampling using deconvolution.

As shown in Table 6, when we remove the hybrid-scale channel attention module and spatial attention module, the OA of the model decreases by 0.12% and 0.05% in the quantitative analysis, respectively. That demonstrates the importance of the attention module for model performance. When we use bilinear interpolation upsampling or nearest interpolation upsampling, there is a substantial degradation in the performance of the model. The OA of HA-Net-T5 decreases by only 0.04%, mainly because upsampling is learnable when using deconvolution. Bilinear interpolation upsampling and nearest interpolation upsampling introduce artifacts that lead to degradation of accuracy.

**Table 6.** Ablation experiments of attention blocks and different upsampling methods.

| Methods | OA | Recall | MIoU | TWR | FWR | *p*-Value (MIoU) |
|---|---|---|---|---|---|---|
| HA-Net | **98.88%** | 98.03% | **97.38%** | **98.24%** | **1.76%** | - |
| HA-Net-T1 | 98.76% | 97.73% | 97.10% | 98.15% | 1.85% | $3.09 \times 10^{-4}$ |
| HA-Net-T2 | 98.83% | 97.98% | 97.27% | 98.13% | 1.87% | $6.61 \times 10^{-4}$ |
| HA-Net-T3 | 98.81% | **98.18%** | 97.21% | 97.86% | 2.14% | $2.31 \times 10^{-4}$ |
| HA-Net-T4 | 98.69% | 97.62% | 96.94% | 98.03% | 1.97% | $4.26 \times 10^{-8}$ |
| HA-Net-T5 | 98.84% | 97.91% | 97.29% | **98.24%** | **1.76%** | $2.60 \times 10^{-4}$ |

*4.4. Experimental Results on the Landsat-8 Dataset*

Currently, a large number of remote sensing image analyses are based on the Landsat-8 remote sensing satellite. Since both the Landsat-8 dataset and the Google dataset consist of RGB images (with the same dimensionality), we can transfer the optimal parameters saved on the Google dataset directly to the Landsat-8 dataset, which not only validates the robustness of HA-Net but also demonstrates that it can be used as a foundation for other lake water body extraction researches.

The visualization results and quantitative results on the Landsat-8 dataset are shown in Figure 11 and Table 7, respectively, where our model obtained the optimal segmentation results. MSLWENet, Pre_DeepLab V3+, and MWFE may lose a large amount of information due to the presence of dilated convolutions in the atrous spatial pyramid pooling (ASPP) structure, which leads to substantial performance degradation during the transfer process. The Pyramid Scene Parsing Module in Pre_PSPNet can increase the ability to understand the features of lakes and avoid a large number of misclassifications. On the Landsat-8 dataset, HA-Net has clear boundaries compared to the segmentation results of other advanced models, which can correctly classify the boundary pixels of the lake water body. As mentioned above, in Figure 11(5), MSLWENet, Pre_DeepLab V3+, and MWFE

have a large number of mis-segmentations due to the presence of ASPP structures. In the quantitative comparison results, although the difference in OA between HA-Net and Pre_PSPNet is only 0.94%, HA-Net has improved 2.93% in TWR compared to Pre_PSPNet. Recall, however, is slightly lower than Pre_PSPNet. All other advanced models have a significant degradation of OA.

**Table 7.** The quantitative results of comparing the proposed method against other methods on the Landsat-8 dataset.

| Methods | OA | Recall | MIoU | TWR | FWR | *p*-Value (MIoU) |
|---|---|---|---|---|---|---|
| HA-Net | **97.28%** | 97.69% | **94.15%** | **94.42%** | **5.58%** | - |
| MSLWENet | 93.76% | 94.34% | 86.99% | 87.30% | 12.70% | $6.35 \times 10^{-14}$ |
| MWEN | 91.01% | 94.27% | 81.48% | 78.95% | 21.05% | $1.32 \times 10^{-15}$ |
| SR-segnet | 92.67% | 98.10% | 84.55% | 80.52% | 19.48% | $1.18 \times 10^{-14}$ |
| Pre_DeepLab V3+ | 91.90% | 96.80% | 83.07% | 79.37% | 20.63% | $8.06 \times 10^{-16}$ |
| Pre_PSPNet | 96.34% | 97.86% | 92.17% | 91.49% | 8.51% | $7.67 \times 10^{-10}$ |
| Pre_Unet | 92.61% | **98.34%** | 84.40% | 80.12% | 19.88% | $4.69 \times 10^{-15}$ |

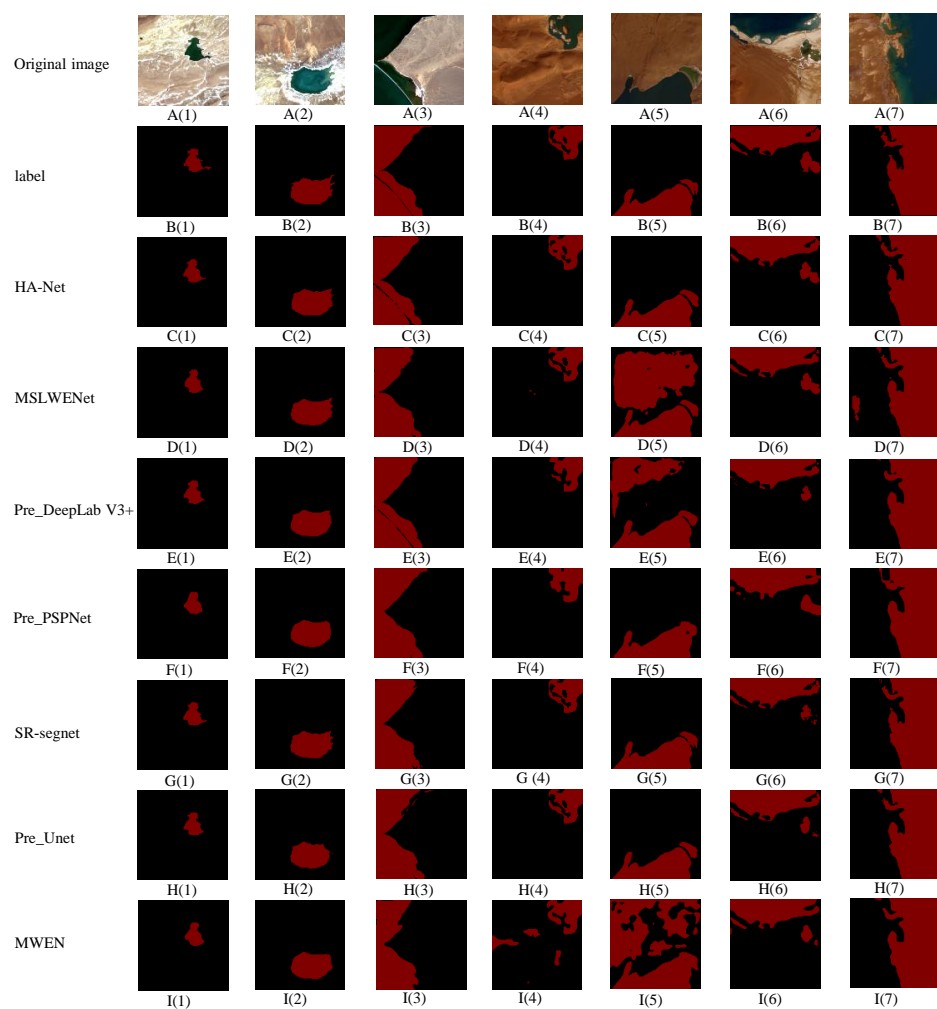

**Figure 11.** The visualization results of comparing the proposed method against other methods on the Landsat-8 dataset. (**A(1)**–**A(7)**) are the raw images in the landsat-8 dataset; (**B(1)**–**B(7)**) are the ground truths to which the raw images correspond; (**C(1)**–**C(7)**), (**D(1)**–**D(7)**), (**E(1)**–**E(7)**), (**F(1)**–**F(7)**), (**G(1)**–**G(7)**), (**H(1)**–**H(7)**), and (**I(1)**–**I(7)**) are the extracted results of HA-Net, MSLWENet, Pre_DeepLab V3+, Pre_PSPNet, SR-segnet, Pre_Unet, and MWEN, respectively.

## 5. Discussion

In this paper, compared to Pre_DeepLab V3+ and Pre_PSPNet for natural image segmentation, Pre_Unet for medical image segmentation, MSLWENet, SR-segnet, and MWFE for water body extraction, our proposed model achieves the best performance on both the Google dataset and Landsat-8 dataset. More importantly, the HA-Net achieves an MIoU of 97.38%, which is a 1.04% improvement over MSLWENet, but reduces the training time by about 100s at each epoch. The Recall and TWR have increased by 0.36% and 0.77% when compared to MSLWENet, despite the OA improvement not being substantial. Since Pre_DeepLab V3+ and Pre_PSPNet use pre-trained weights, their segmentation performance is comparable to the MSLWENet.

In the small lake and noisy regions, it can be seen from the visualization results that our method has comparable performance with MSLWENet. Meanwhile, as shown by the quantitative analysis, OA and MIoU improved by 0.37% and 0.75%, respectively. HA-Net has great segmentation performance on the fuzzy boundaries in the boundary regions. In particular, we discuss the effect of various encoders on classification accuracy. As the depth of ResNet increases, the training time of the model becomes longer, but there is a slight decrease in its performance. This is mainly because the number of parameters in the deeper models becomes larger, making it slightly overfit. The performance of both the shallow network represented by VGG and the densely connected network represented by DenseNet decreases significantly, thus proving the superiority of ResNet, which is mainly due to the fact that the dilated convolution expands the receptive field and allows more useful information to be extracted. EfficientNet takes full consideration of depth, width, and resolution of the input image, although achieving OA of 98.83%, the training time is greatly increased. Finally, we conduct a comparison experiment between the hybrid-scale attention block and the pixelshuffle upsampling convolution block to demonstrate the superiority of HA-Net. When transferring the optimal weights to the Landsat-8 dataset, our model has more robustness compared to other advanced models. On the other hand, we also perform a common non-parametric test method to assess whether our performance improvements are statistically significant in terms of MIoU metrics compared to other current state-of-the-art methods, resulting in a p-value score. From the *p*-values shown in Tables 1 and 3–7, it is clear that there is a statistically significant improvement in the MIoU metric for our method at the 5% level (all *p*-values are less than 0.05).

## 6. Conclusions

The Tibetan Plateau is the region with the largest number of lakes and one of the two most densely distributed lake regions in China. Although Wang et al. [24] used depthwise separable convolution to limit the model complexity, its training time is not optimized.

In this paper, a new semantic segmentation model based on a convolutional neural network for lake water body extraction from remote sensing images is proposed. In the downsampling stage, we propose a two-branch encoder, which uses a shallower encoder, ResNet-34, to extract features and a feature pyramid network to fuse features at different scales. It not only handles the multiscale problem of the target but also greatly reduces the model complexity and training time of the model. The hybrid-scale attention block is used to adaptively weigh the feature maps generated by the feature pyramid network in the channel and spatial dimensions, thereby suppressing the interference of noise. In the upsampling stage, the pixelshuffle convolution upsample block is used to recover the resolution of the feature maps and refine the segmentation boundaries. Compared with MSLWENet, our model not only can correctly extract small lake water bodies and suppress the interference of noise but also can improve the accuracy of target segmentation boundaries.

A transformer is a new research hotspot in the field of semantic segmentation of natural images, which treats it as a sequence-to-sequence prediction task to replace convolutional neural networks. Transformer performs global context modeling for each layer and achieves semantic segmentation by combining it with a simple decoder. Our subsequent

work will mainly focus on the extraction of the lake water bodies using the transformer method and verifying its feasibility.

**Author Contributions:** methodology, X.G.; data curation, X.G.; writing, original draft preparation, X.G.; writing, review and editing, Z.W.; supervision, Z.W.; project administration, Y.Z.; funding acquisition, Z.W. All authors read and agreed to the published version of the manuscript.

**Funding:** This research was funded by the National Natural Science Foundation of China (Grant No. 61201421), the National cryosphere desert data center (Grant No. E01Z7902), and the Capability improvement project for cryosphere desert data center of the Chinese Academy of Sciences (Grant No. Y9298302).

**Institutional Review Board Statement:** Not applicable.

**Informed Consent Statement:** Not applicable.

**Data Availability Statement:** The data presented in this study are available on request from the corresponding author.

**Acknowledgments:** We would like to thank all the people who helped and supported our research, especially Minzhe Xu, Yikun Ma, and Zhongxin Cheng and the staff at CAS.

**Conflicts of Interest:** The authors declare no conflict of interest.

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
