# Peer review of "HA-Net: A Lake Water Body Extraction Network Based on Hybrid-Scale Attention and Transfer Learning"

_remotesensing, doi:10.3390/rs13204121_

Round 1

Reviewer 1 Report

I accept the manuscript in its current form. Congratulations for the paper. It is very clear and well explained. My only comment is to try to be more precise in the conclusions by highlighting the accuracy of the model.

Author Response

Point 1: I accept the manuscript in its current form. Congratulations for the paper. It is very clear and well explained.

Response 1: Thank you very much for your recognition of our manuscript. We have revised the conclusions of the model to make them more precise based on your comments.

Point 2: My only comment is to try to be more precise in the conclusions by highlighting the accuracy of the model.

Response 2: In the abstract section, we have replaced the Overall Accuracy (OA) with Mean Intersection over Union (MIoU), which has the largest performance improvement, and presented the overall accuracy of HA-Net with Recall, True Water Rate (TWR), and False Water Rate (FWR) of 98.88%, 98.03%, 98.24%, and 1.76% respectively.

In the discussion section, we have similarly replaced OA with MIoU and discussed the Recall changes.

In addition, we calculated the significance statistics for the different methods of the MIoU metric based on other reviewer comments.

Reviewer 2 Report

Attached.

Reviewer 3 Report

Dear authors, 

After reviewing your manuscrip, I found your finds are a bit superior against similar approaches. I am wondering if that come from selected images not from method's performance. 

I share with you the following observations:

Did you find the metrics values using all images from dataset, I mean, 610 from Google and 841 from Landsat8? Or just the 5 google and 7 landsat images?

Although your results showed superior performance compared against other nets, these are very slight, almost negligible. Maybe I would think that if you retire some images with no good performance, you could improve the performance of the net you choose, isn't it?

Why you do not evaluate the boundary's performance on the same image set used for small lake and noisy regions? If you did it, how was the perfomance?

References 22 & 23 are the same
Some mispelled words such as knrnel, snalyses

Best regards, 

Round 2

Reviewer 3 Report

Dear authors, 

I felt satisfied with your answers to my inquiries, so I consider it is ready for publication as it is.

Best regards,